

# Resolution-agnostic tissue segmentation in whole-slide histopathology images with convolutional neural networks

Péter Bándi[1], Maschenka Balkenhol[1], Bram van Ginneken[2], Jeroen van der Laak[1] and Geert Litjens[1]

[1] Department of Pathology, Radboud University Medical Center, Nijmegen, The Netherlands
[2] Department of Radiology and Nuclear Medicine, Radboud University Medical Center, Nijmegen, The Netherlands

## ABSTRACT

Modern pathology diagnostics is being driven toward large scale digitization of microscopic tissue sections. A prerequisite for its safe implementation is the guarantee that all tissue present on a glass slide can also be found back in the digital image. Whole-slide scanners perform a tissue segmentation in a low resolution overview image to prevent inefficient high-resolution scanning of empty background areas. However, currently applied algorithms can fail in detecting all tissue regions. In this study, we developed convolutional neural networks to distinguish tissue from background. We collected 100 whole-slide images of 10 tissue samples—staining categories from five medical centers for development and testing. Additionally, eight more images of eight unfamiliar categories were collected for testing only. We compared our fully-convolutional neural networks to three traditional methods on a range of resolution levels using Dice score and sensitivity. We also tested whether a single neural network can perform equivalently to multiple networks, each specialized in a single resolution. Overall, our solutions outperformed the traditional methods on all the tested resolutions. The resolution-agnostic network achieved average Dice scores between 0.97 and 0.98 across the tested resolution levels, only 0.0069 less than the resolution-specific networks. Finally, its excellent generalization performance was demonstrated by achieving averages of 0.98 Dice score and 0.97 sensitivity on the eight unfamiliar images. A future study should test this network prospectively.

Corresponding author
Péter Bándi,
Peter.Bandi@radboudumc.nl

## INTRODUCTION

Digital pathology is opening new avenues for pathologists. Straightforward archiving, remote diagnostics, and application of image analysis to improve the efficiency of the diagnostic process are among the most commonly mentioned advantages of digital pathology (*Snead et al., 2016*).

After fixation (typically with formalin) and paraffin embedding, a couple of micrometers thin slices are cut from the tissue specimens and placed on glass slides.
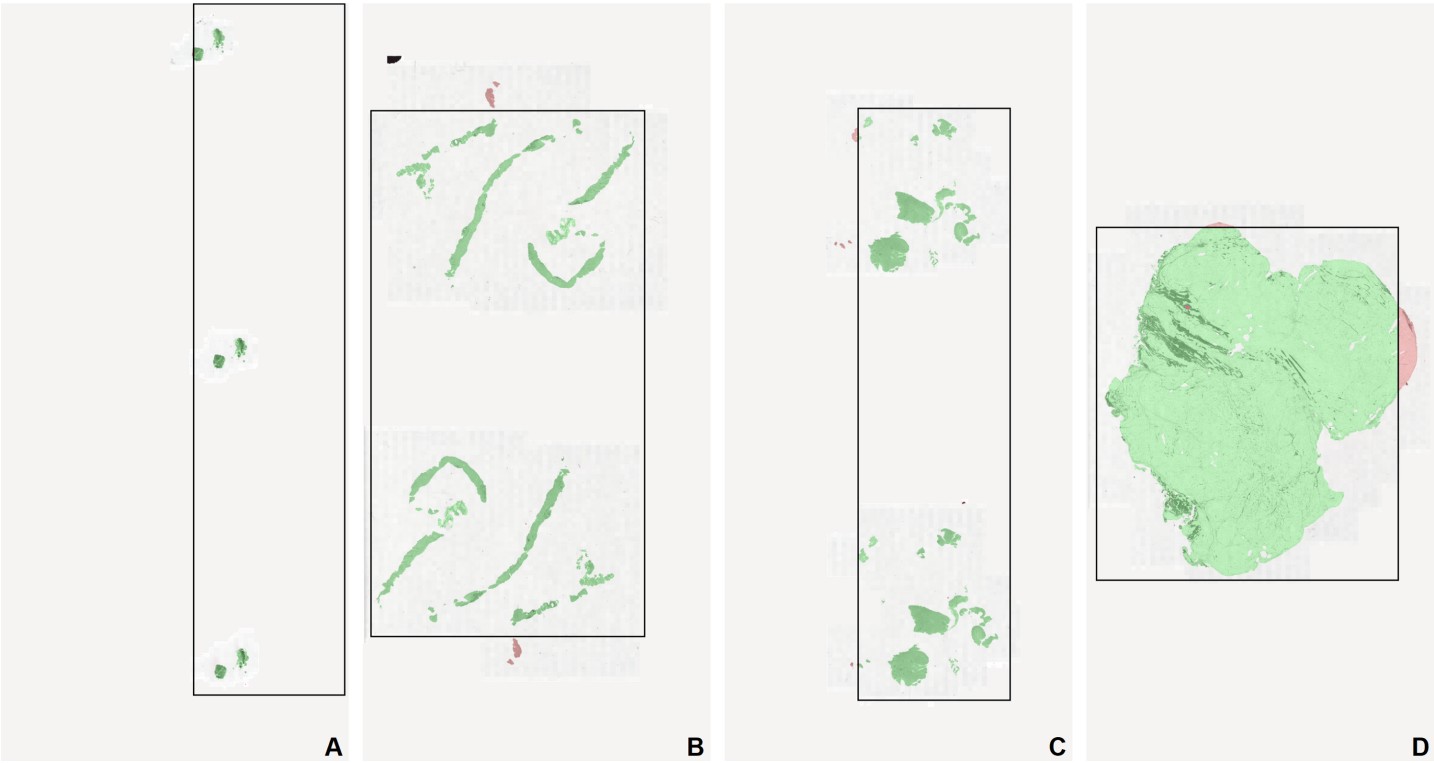

**Figure 1 Examples of failed tissue detections by a whole-slide scanner.** The four glass slides (A–D) were first scanned in the Cannizzaro Hospital (Catania, Italy) with a Leica Aperio AT2 whole-slide scanner, then re-scanned for this study at the Radboud UMC, in Nijmegen with a 3DHistech P250 Flash II whole-slide scanner, forcing the scanner to fully scan the entire slide. Color overlays: tissue scanned by the Leica scanner is green, missed tissue is red. The black rectangles outline the regions that were selected for scanning by the tissue detector algorithm of the Leica scanner.

These slides are then stained to highlight the required features specific to the intended application. Traditionally, the pathologist analyzes the slides through microscopic assessment. Whole-slide scanners are increasingly used to digitize glass slides containing tissue sections, producing so-called whole-slide images (WSI).

Having digital WSIs enables a digital workflow, replacing the physical glass slides. It also opens the possibility of processing WSIs to aid the pathologist; future developments could even lead to a completely automated assessment procedure.

However, the digital workflow comes with its own challenges. Whole-slide scanners aim to detect all tissue on the glass slides to decide which areas to scan, and to determine the optimal focus depth for those areas. It is well known that scanners sometimes fail in detecting all tissue regions, for example, due to the tissue type (e.g., fatty tissue is difficult to detect because of the relatively large transparent areas within the individual fat cells) or weak staining (e.g., immunohistochemistry). Missed tissue regions cause large risks for quality of diagnostics, for instance in detection of cancerous regions in lymph nodes. Examples of failed tissue detections are shown in Fig. 1.

Unfortunately, there is no way to recover from errors made in tissue detection by slide scanners in later steps of the digital pathology workflow in an automated setting.

The easiest solution—to scan the entire slide—is not feasible in clinical practice as it would increase the scan time and file size beyond reasonable limits. Another option is to manually check each and every scanned WSI. This, however, quickly becomes an expensive and time-consuming solution in laboratories with fully digital pathology workflow where hundreds of slides are scanned daily.

The identification of tissue areas is also relevant for already-obtained WSIs. By highlighting the tissue, one can prevent unnecessary processing of large parts of the WSI by subsequent algorithms. In the CAMELYON17 challenge dataset (*Bándi et al., 2018*), for example, only 22% of the total area of the WSIs is tissue.

Prior work has already been done on this topic. The most straightforward method to identify tissue on the white background is to threshold the grayscale version of the image (*Hart et al., 2018*), the individual color channels (*Yu et al., 2016*), or the optical density of the RGB channels (*Xia et al., 2018*) of the image at a predefined value. In other cases the WSIs were divided into a uniform image patch grid and the grayscale image was first thresholded then the resulting binary tissue map was use to discard image patches that does not contain enough tissue (*Liu et al., 2019*; *Gertych et al., 2019*; *Coudray et al., 2018*; *Liu et al., 2017*).

In the CAMELYON17 challenge (*Bándi et al., 2018*), most of the participants used Otsu's adaptive thresholding (*Otsu, 1979*) method to identify the tissue areas on the WSIs for effective training of their algorithms and the algorithm has been used for the same purpose in other studies too (*Campanella et al., 2019*; *Nirschl et al., 2018*; *Xu, Park & Hwang, 2019*). Others complemented the method with region growing from the edge of the binary images generated by Otsu's threshold (*Vanderbeck et al., 2014*).

Furthermore, some groups have already tried to design methods to improve the tissue detection in scanners. *Bug, Feuerhake & Merhof (2015)* used a method called foreground extraction from structure information (FESI) based on global thresholding at the mean value of the Gaussian-blurred Laplacian of the grayscale image; the result is subsequently refined via flood filling from identified background points. *Hiary, Alomari & Chaudhary (2013)* built a different algorithm based on k-means clustering using pixel intensity, color, and texture features.

Thresholding at a predefined value and Otsu's method are very straightforward but difficult to adapt to more complex stains or tissues. FESI is composed of traditional image processing steps with many predetermined constants fine-tuned on a dataset with limited variation. As such, both do not generalize well to the variation of WSIs encountered in clinical practice. We propose a solution based on convolutional neural networks (CNNs) (*Krizhevsky, Sutskever & Hinton, 2012*). CNNs have already been shown to excel in image segmentation tasks (*Ehteshami Bejnordi et al., 2017*; *Setio et al., 2017*; *Bándi et al., 2018*). To ensure that our solution performs well on a broad spectrum of images, we collected 100 WSIs of 10 different tissue-staining categories from a wide range of sources for development and testing—allocated to categories based on unique tissue and staining combinations. Moreover, eight categorically-unique WSIs were collected for additional testing.

In this paper, we compared our proposed fully-convolutional deep learning approach to thresholding at a predefined value, Otsu's adaptive thresholding, and the FESI method from Bug et al. with respect to tissue segmentation accuracy. Additionally, we show that a single fully convolutional neural network (FCNN) that is developed for a range of image resolutions can perform comparably to multiple resolution-specific FCNNs. Finally, we test how well our method generalizes on tissue types and stainings that were not part of the development set.

## MATERIALS

### Whole-slide images

The slides were collected from five medical centers in the Netherlands and Germany: 58 from Radboud University Medical Center in Nijmegen, the Netherlands (RUMC), 10 from Canisius-Wilhelmina Hospital in Nijmegen, the Netherlands (CWZ), 20 from Laboratory of Pathology East-Netherlands in Hengelo, the Netherlands (LPON), 10 from Heidelberg University Hospital in Heidelberg, Germany (HUH), and 20 from Hannover Medical School in Hannover, Germany (HMS). The collection of the data was approved by the local ethics committee (Commissie Mensgebonden Onderzoek regio Arnhem—Nijmegen) under 2019-5161; the need for informed consent was waived.

The glass slides were digitized with whole-slide scanners, resulting in WSIs. The WSIs contained multiple resolution levels, with approximately $1 \times 10^5$ by $2 \times 10^5$ pixels at the highest resolution level. Each consecutive resolution level doubled the pixel size in both directions and halved the pixel count in each dimension. The typical file size of a WSI was three GB, but it varied greatly depending on the scanner, the scanning settings including the pixel spacing, and tissue content of the image. The vendor-specific image formats were anonymized and converted to standard multi-resolution TIFF image files. For a description of the file format, see http://openslide.org/formats/generic-tiff.

We assembled two datasets. We refer to them as the *development dataset* and the *dissimilar dataset*. The development dataset was composed of 100 WSIs in total and was split into training, validation, and testing subsets. The training set was used to optimize network weights, and the validation set to optimize hyperparameters. The test set was untouched during algorithm development and only used for the final evaluation. The dissimilar dataset was composed of eight images and was used for testing only. The two datasets are described in the following sections.

#### Development dataset

A wide variety of glass slides were collected for training and testing in order to ensure that the trained network generalizes well over different tissue types and stains. We also accounted for the differences between staining protocols and scanners by including glass slides from five different medical centers, and scanned by five different scanners.

Six different tissue types were included in the dataset: breast, lymph node, kidney, tongue, rectum, and lung tissue. The slides were stained with six different stains: hematoxylin and eosin (H&E), Sirius Red, Periodic Acid-Schiff (PAS), cytokeratin AE1/AE3 (AE1AE3), Ki-67, and a cocktail of cytokeratin 8 and cytokeratin 18 (CK8-18).

**Table 1 The 10 different tissue-staining categories of the development dataset and the highest resolution pixel sizes are displayed.** The dataset was randomly divided into training, validation, and testing sets with a 5, 2, 3 distribution, respectively. See the text for center and staining abbreviations.

| Tissue | Staining | Center | Pixel size (μm) | Count | Average size (GB) |
|---|---|---|---|---|---|
| Breast | H&E | RUMC | 0.2431 | 10 | 3.6161 |
| Lymph node | CK8-18 | LPON | 0.2500 | 10 | 0.6626 |
| Lymph node | H&E | CWZ | 0.2431 | 10 | 2.0881 |
| Kidney | PAS | HMS | 0.5034 | 10 | 0.2405 |
| Kidney | Sirius Red | HMS | 0.2525 | 10 | 2.3294 |
| Lung | H&E | HUH | 0.2278 | 10 | 3.2345 |
| Rectum | H&E | LPON | 0.2275 | 10 | 4.4282 |
| Tongue | AE1AE3 | RUMC | 0.2431 | 10 | 4.6374 |
| Tongue | H&E | RUMC | 0.2431 | 10 | 4.3225 |
| Tongue | Ki-67 | RUMC | 0.2431 | 10 | 4.5081 |

Hematoxylin and eosin is the most commonly used histochemical staining in pathology. Haematoxylin stains basophilic structures, and eosine stains the acidophilic structures. In practice, this results in an image of several shades of pink, with dark blue nuclei. PAS staining is usually used for kidney tissue as it provides a red-purple reaction product with glycolgroups in, for example, basal membranes in the tissue. Sirius Red is a histochemical staining for muscle fibers and collagen I and III. It stains collagen red, and cytoplasm and muscle fibers yellow. The immunohistochemical stainings (AE1/AE2, Ki-67, and CK8-18) make use of the principle of specific antigen-antibody binding. We used 3,3-Diaminobenzidine to stain, which results in a brown color at the targeted antigens. Haematoxylin was used as counter staining for the immunohistochemical stainings, resulting in blue nuclei.

The slides from RUMC and CWZ were scanned in the RUMC with a 3DHistech P250 Flash II whole-slide scanner with a pixel size of 0.2431 μm; LPON used a Philips IntelliSite Ultra Fast Scanner with 0.25 μm pixel size and a Hamamatsu NanoZoomer C9600-13 scanner with 0.2275 μm pixel size; HUH used a Hamamatsu NanoZoomer C9600-12 scanner with 0.2278 μm pixel size; and HMS used a Leica Aperio AT2 whole-slide scanner at 0.5034 and 0.2525 μm pixel sizes.

The unique combinations of tissue type and staining yielded 10 different categories. Having included 10 WSIs from each category, our dataset contained a total of 100 WSIs. The complete list of WSIs is shown in Table 1. Each class of WSIs in this dataset was divided randomly into training, validation, and testing sets with a 5, 2, 3 distribution, respectively, yielding evenly distributed categories in all subsets. Examples of all 10 categories are shown on Fig. 2.

### Dissimilar dataset

In order to test how well our proposed method performs on tissue types and stainings that it had not been trained on, we collected eight more images from RUMC. The images were scanned with a 3DHistech P250 Flash II whole-slide scanner with a pixel size of 0.2431 μm.

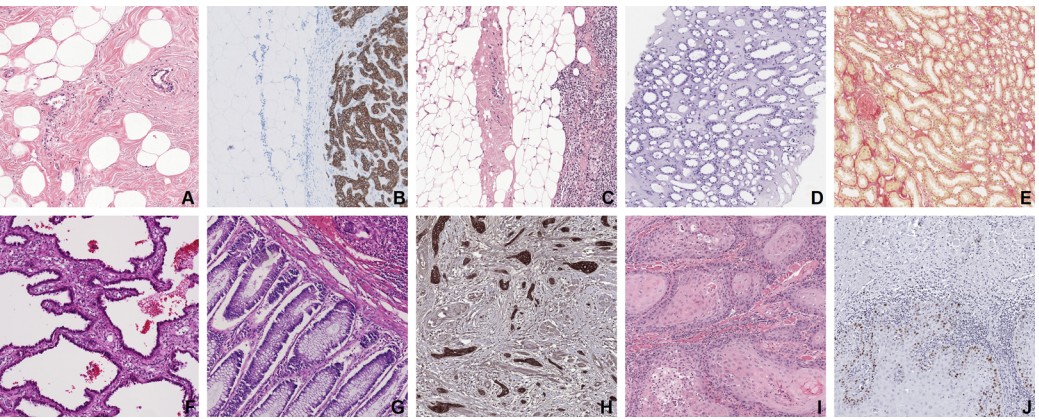

**Figure 2 Examples of all 10 tissue—staining categories in the development dataset.** (A) Breast—H&E, (B) lymph node—CK8-18, (C) lymph node—H&E, (D) kidney—PAS, (E) kidney—Sirius Red, (F) lung—H&E, (G): rectum—H&E, (H) tongue—AE1AE3, (I) tongue—H&E, (J) tongue—Ki-67. The field of view is 840 × 840 μm.               

**Table 2 The eight images of the dissimilar dataset that was used only for testing.** None of the tissue-staining categories had been included in the development dataset.

| Tissue | Staining | Center | Pixel size (μm) | Count | Size (GB) |
|--------|----------|--------|-----------------|-------|-----------|
| Aorta | Alcian Blue | RUMC | 0.2431 | 1 | 3.9859 |
| Brain | Alcian Blue | RUMC | 0.2431 | 1 | 5.4795 |
| Cornea | Grocott | RUMC | 0.2431 | 1 | 2.8011 |
| Kidney | CAB | RUMC | 0.2431 | 1 | 1.9790 |
| Lung | Perls | RUMC | 0.2431 | 1 | 5.1517 |
| Skin | Perls | RUMC | 0.2431 | 1 | 4.5908 |
| Skin | Von Kossa | RUMC | 0.2431 | 1 | 2.8210 |
| Uterus | Von Kossa | RUMC | 0.2431 | 1 | 3.0766 |

This dataset contained lung, cornea, aorta, brain, skin, uterus, and kidney tissue samples. The tissues were stained with Grocott, Alcian Blue, Von Kossa, Perls, and Chromotrope Aniline Blue (CAB) stains. Only lung and kidney tissues were present in both of the datasets, while the stains are non-overlapping.

For the complete list of WSIs, we refer to Table 2. All eight images in this dataset were used for testing only. Samples of all eight images are shown on Fig. 3.

## Tissue annotations

In all of the included WSIs, the tissue areas were carefully outlined by medical students on the highest resolution, following the outer edge of the tissue regions as closely as possible. If the tissue fell apart to multiple disjunct pieces, for example, due to preparation or staining, the students were to annotate them with one enclosing polygon. Tissue areas further apart than 50 μm were annotated as separate regions. Subsequently, annotations were checked by a pathology resident (M.B.) to verify that no clinically relevant areas were missed.

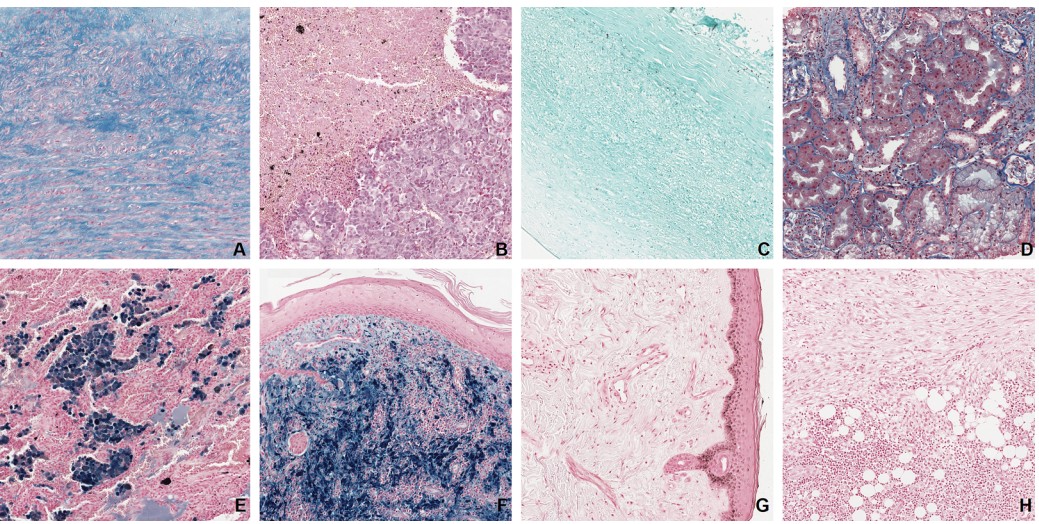

**Figure 3 Examples of all eight tissue-staining categories in the dissimilar dataset.** (A) Aorta—Alcian Blue, (B) brain—Alcian Blue, (C) cornea—Grocott, (D) kidney—CAB, (E) lung—Perls, (F) skin—Perls, (G) skin—Von Kossa, (H) uterus—Von Kossa. The field of view is 840 × 840 μm.

Holes in the tissue, larger than 250 μm in diameter—possibly tearing as a result of the processing—or anatomical structures (e.g., ducts and vessels) were annotated as *background*.

Some slides also contained non-tissue artifacts like air bubbles, edges of the cover slip, stain residue, markings on the glass slide, and tissue debris. While these regions should be classified as background, they have very different characteristics from the homogeneous white background areas and were proven to be difficult to identify accurately as such. In order to be able to sample specifically from these background areas during the training of the CNNs, we also annotated the aforementioned artifacts. Cover slip edge, air bubbles at the edge, and glass slide markers were annotated as *edge artifacts*, and every other artifact including stain residue, tissue debris, and inner air bubbles were annotated as *inner artifacts*.

All together, there were four annotation groups: *tissue*, *background*, *edge artifacts*, and *inner artifacts*. An example WSI annotation is shown on Fig. 4.

Finally, to reduce variability between annotators, we removed every annotated hole less than 250 μm in diameter, every tissue region that was less than 250 μm in diameter, and merged annotated tissue areas that were within 50 μm.

We used ASAP software for annotating the images. ASAP is an open source software that is available at https://github.com/computationalpathologygroup/ASAP.

## Sampling masks

To train the proposed CNNs, we needed to sample patches from the annotated areas. To facilitate this, the annotations were converted to mask images that had a single class label for every pixel. For training and validation purposes, all four annotation groups (i.e., *tissue*, *background*, *edge artifacts*, and *inner artifacts*) were given distinct labels.

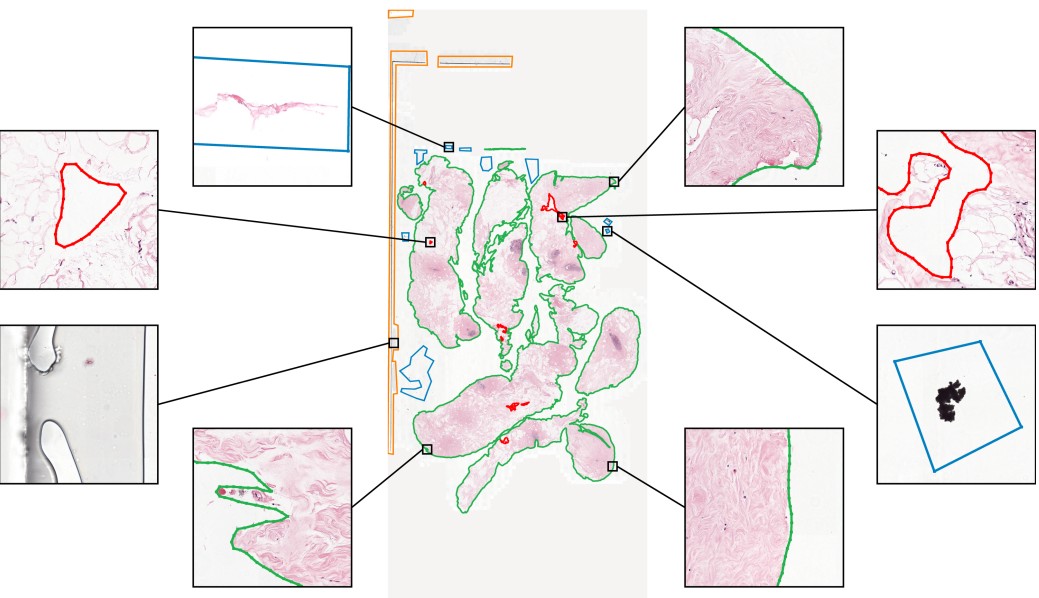

**Figure 4 An example WSI annotation with enlarged sample areas.** Colors: tissue annotation is green, background annotation is red, edge artifacts annotation is orange, inner artifacts annotation is blue.

All non-annotated pixels were given the *background* label value. This allowed us to specifically direct patch sampling to the more difficult classes, such as artifacts.

In addition, pixels close to the edge of the tissue are a natural source of segmentation ambiguity. As such, we decided to also give edge pixels distinct class labels. The pixels labeled as *background* and within 125 pixels to any tissue area were assigned the *external-margin* label, while pixels labeled as *tissue* and closer than 125 pixels to any *background* area were assigned the *internal-margin* label. This yielded masks with six labels for training and validation.

For training the CNNs, we sampled patches from specific resolution levels of the WSIs. The margins of 125 pixels was 62.5, 250.0, and 1,000.0 μm at resolution levels with 0.5, 2.0, and 8.0 μm pixel spacing, respectively.

As we trained the CNNs to produce binary predictions (tissue or background), the six labels were only used to control the sampling ratio and subsequently converted to a "tissue" or "non-tissue" label for network training. For a complete listing of labels and the mapping of the labels to "tissue" or "non-tissue" flags, we refer to Table 3.

## METHODS

### Thresholding at a predefined value

First, the WSIs were converted to grayscale by averaging the RGB channels. The resulting images were thresholded at the fixed $i = 217$ value. Subsequently, as *Gertych et al. (2019)* we refined the tissue mask by hole filling and morphological closing. The $i$ threshold was determined by calculating the tissue masks of the validation subset of the

**Table 3 Labels of the six class masks that were generated from the manual annotations of the WSIs.** The masks were used to select the positions of image patches in the WSIs to sample for training of the CNNs.

| Name | Tissue | Description |
|---|---|---|
| Edge | No | Artifacts on the edge of the glass slide |
| Artifacts | No | Artifacts inside the glass slide |
| Background | No | Background area without artifacts |
| External-margin | No | Background area close to the tissue border |
| Internal-margin | Yes | Tissue area close to the tissue border |
| Tissue | Yes | Tissue area |

development set and selecting the threshold value that achieved the highest average Dice score (*Dice, 1945*).

## Otsu's adaptive threshold

Otsu's method (*Otsu, 1979*) is a clustering-based image thresholding algorithm. The algorithm assumes that the image contains two classes of pixels following bi-modal histogram (tissue pixels and background pixels). It calculates the optimum threshold separating the two classes so that their combined intra-class variance is minimal. It has been widely used in image analysis applications and digital histopathology (*Bándi et al., 2018*; *Azevedo Tosta, Neves & do Nascimento, 2017*; *Campanella et al., 2019*; *Nirschl et al., 2018*; *Xu, Park & Hwang, 2019*; *Vanderbeck et al., 2014*). For applying Otsu's method the WSIs were first converted to grayscale by averaging the red, green, and blue channels.

## Foreground extraction from structure information

Foreground extraction from structure information (*Bug, Feuerhake & Merhof, 2015*) uses an edge detector to get an initial separation of the structured tissue and homogeneous background areas. After further refinement of the initial selection by median blurring and morphological opening, a flood filling is started from the point furthest away from any tissue area to select the background, thus filling the holes in the tissue areas.

Finally, the small tissue regions were removed by calculating a distance transformation on the current tissue mask and iterating through the maxima. In each step, the point with the maximal distance value is taken as a seed point. If its value is larger than 100 or is within 100 pixels from any previously accepted seed point, it is accepted and its containing region is marked as tissue; otherwise, it is discarded. The steps of the algorithm are shown in Fig. 5.

## Convolutional neural networks

In this section, we discuss the architecture of our CNNs and the training process.

### Architecture

It has been shown that FCNNs can achieve excellent performance in WSI processing tasks (*Ehteshami Bejnordi et al., 2017, 2018*; *Bándi et al., 2018*; *Nagpal et al., 2019*; *Liu et al., 2019*), and that FCNNs can reach similar Dice score compared to U-Net architectures
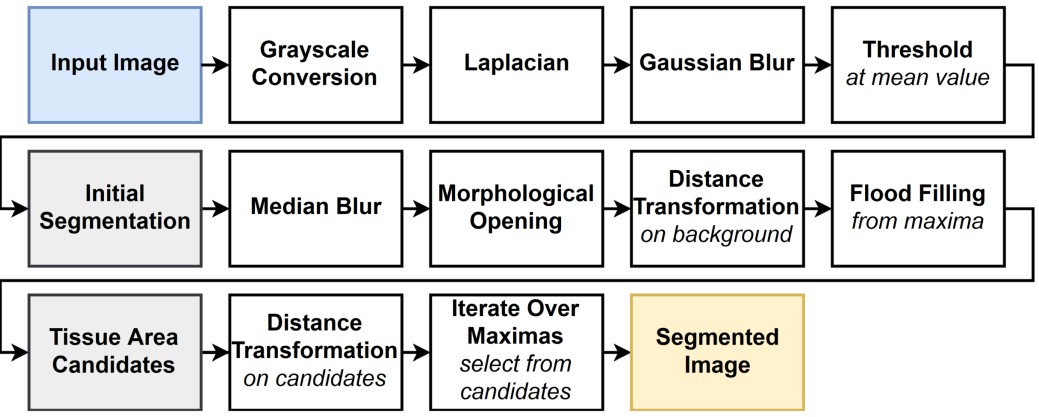

**Figure 5  Flowchart of the FESI algorithm.** The filled boxes represent images, while the non filled boxes are algorithmic steps.

**Table 4  Network architecture (all our CNNs had the same architecture).**

| Layer | Type | Filter size | Stride | Filter count | Activation |
|---|---|---|---|---|---|
| 0 | CONV | 5 × 5 | 1 | 16 | ReLU |
| 1 | POOL | 2 × 2 | 2 | | |
| 2 | CONV | 5 × 5 | 1 | 32 | ReLU |
| 3 | POOL | 2 × 2 | 2 | | |
| 4 | CONV | 3 × 3 | 1 | 64 | ReLU |
| 5 | POOL | 2 × 2 | 2 | | |
| 6 | CONV | 3 × 3 | 1 | 64 | ReLU |
| 7 | CONV | 3 × 3 | 1 | 1,024 | ReLU |
| 8 | CONV | 1 × 1 | 1 | 512 | ReLU |
| 9 | CONV | 1 × 1 | 1 | 2 | softmax |

(*Ronneberger, Fischer & Brox, 2015*) in tissue segmentation (*Bándi et al., 2017*). We used FCNN architecture in this study because of its similar performance in tissue segmentation at a significantly lower computational cost.

Our fully convolutional neural network (FCNN) (*Shelhamer, Long & Darrell, 2017*) consisted of seven convolutional layers with ReLU (*Maas, Hannun & Ng, 2013*) activation function in the first six convolutional layers and softmax in the last one. Max pooling was inserted after each of the first three convolutional layers to reduce the memory requirements of the network. Table 4 describes the complete network architecture.

### Training

The networks were initialized with the He (*He et al., 2015*) method and the weights were updated using the Adam optimizer (*Kingma & Ba, 2014*). We used categorical cross entropy as the loss function and added L2 regularization with a weight of $\lambda = 10^{-5}$. The $l = 10^{-4}$ initial learning rate was divided by 2 after each four consecutive epochs without improvement, and the training procedures were stopped after 16 consecutive epochs without improvement.

The networks were trained with RGB image patches of 128 × 128 pixels that were randomly sampled from the WSIs during training. Each epoch consisted of 25,600 training and 6,400 validation iterations. In each iteration, we extracted 32 image patches to form a mini-batch. The measured epoch accuracy of the networks was the average accuracy across validation iterations.

### Sampling

The RGB image patches were sampled with uniform distribution over the WSI classes for both training and validation purposes, with a 1, 1, 1, 1, 5, 1 distribution for edge, artifacts, background, external-margin, internal-margin, and tissue areas, respectively.

### Resolution

We trained two types of networks: FCNNs A, B, and C were single-level networks trained with image patches extracted from a single level of the multi-resolution WSIs; FCNN D was a multi-level network trained with image patches extracted from a range of levels. FCNN A, B, and C were trained with image patches from the levels closest to 0.5, 2.0, and 8.0 µm pixel spacing, respectively.

For the training of the FCNN D, the source level was randomly chosen for each loaded image patch, from the levels closest to 0.5, 1.0, 2.0, 4.0, or 8.0 µm pixel spacing. For the sake of simplicity, we refer to the levels with 0.5, 2.0, and 8.0 µm approximate pixel spacing as *level 1*, *level 3*, and *level 5*, respectively.

### Augmentations

We used data augmentation during training to make our networks more robust to data variations encountered in practice. Specifically, we aim to simulate the differences between scanners, stains, and staining protocols. The loaded image patches were subjected to a series of augmentation steps with randomized parameters from pre-defined ranges with uniform distribution for each individual patch.

Our augmentation methods greatly overlap with the published *HSV-Strong* method (*Tellez et al., 2019*). However, we did not use elastic transformation, the individual augmentation methods are executed in different order in our pipeline, and the ranges of the parameters were extended for greater variability. The augmentation pipeline consisted of horizontal mirroring, 90° rotations, scaling, color adjustment in hue-saturation-brightness color space, contrast adjustment, additive Gaussian noise, and Gaussian blur. Table 5 lists the augmentation steps and their parameter ranges.

By using Hue augmentation from the *HSV-Strong* method we made every color possible on the image patches while keeping the distances between the colors unchanged. This way, the trained networks were forced to be independent of the absolute hue value, thus be able to recognize tissue with with unseen stains. Examples of the hue augmentation are shown in Fig. 6.

### Inference

For WSI segmentation, the networks were applied in a fully convolutional fashion to the image (*Shelhamer, Long & Darrell, 2017*). The output of the networks were pixel-wise

**Table 5 Augmentation configuration: each loaded image patch was subjected to a series of augmentation steps in the order listed, with parameters randomly picked from the ranges shown.**

| Order | Augmentation | Parameter | Range |
|---|---|---|---|
| 1 | Mirroring | $f$ | Horizontal, none |
| 2 | Rotation | $\alpha$ | 0°, 90°, 180°, 270° |
| 3 | Scaling | $z$ | (0.75, 1.25) |
| 4 | Hue adjustment | $h$ | (−1.0, 1.0) |
| 5 | Saturation adjustment | $s$ | (−0.25, 0.25) |
| 6 | Brightness adjustment | $b$ | (−0.25, 0.25) |
| 7 | Contrast adjustment | $c$ | (−0.25, 0.25) |
| 8 | Additive Gaussian noise | $\sigma_a$ | (0.0, 0.05) |
| 9 | Gaussian blur | $\sigma_b$ | (0.0, 1.0) |

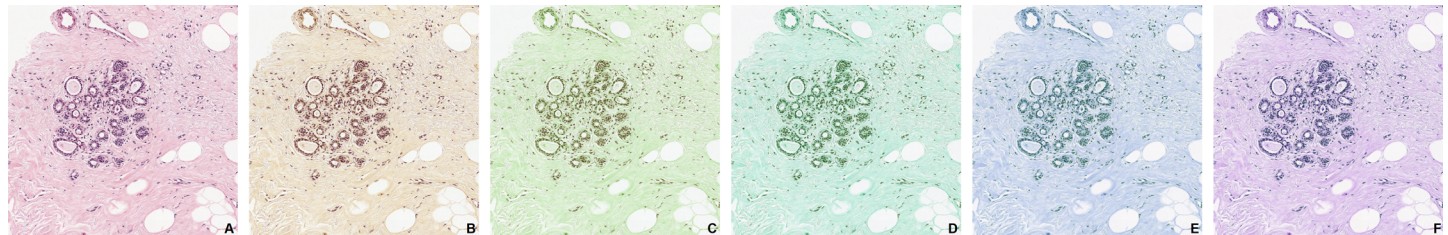

**Figure 6 Examples of hue augmentation.** Form (A–F) adjustments with 0°, 60°, 120°, 180°, 240°, and 300°.

probabilities (range 0.0–1.0) of belonging to tissue. This output was thresholded at $t = 0.8$, resulting in a binary tissue-background mask. The confidence threshold was selected by measuring the average Dice score on the development validation set.

## Post-processing

For fair comparison, we used the same post-processing method on the binary tissue-background masks for all the measurements, including the traditional methods. Since the annotated tissue regions smaller than 250 µm in diameter were removed and the holes smaller than 250 µm were filled in the reference, we removed every disjunctive region from the binary output masks of the algorithms that was smaller than 250 µm in diameter. We then filled every hole in the output tissue regions that was smaller than 250 µm.

## Measurements

We quantitatively compared the algorithms using the Dice score (*Dice, 1945*), comparing the output of the algorithms against the binary reference masks of the 30 images in the testing part of the development dataset.

In addition, we calculated the sensitivity and number of false positives for each slide. We considered a reference tissue area a true positive if at least 80% of its area was detected by the algorithm as tissue; otherwise, the region was counted as a false negative. A region of the algorithm output was counted as a false positive detection if it had no overlap with any of the reference tissue regions.
To test if our FCNN was significantly different from Otsu's method and the FESI algorithm, we applied pairwise *t*-tests with Bonferroni correction using the Dice scores of the algorithms on development test set. Differences with $p < 0.05$ were considered statistically significant.

# RESULTS

## Comparison with traditional methods

First, we compared our FCNN method against the traditional thresholding, Otsu's, and the FESI method. We executed the algorithms on levels 1, 3, and 5. However, we could not execute the FESI method on level 1 due to memory limitations (it required more than 200 GB at this level).

By selecting these resolutions, we are able to provide results which cover a wide range of use-cases—from coarse initial segmentation at low resolution, to detailed segmentation at high resolution. When selecting the level in a WSI with a given approximate pixel spacing, we selected the one with the pixel spacing closest to the target. For example, for 0.5 μm approximate pixel spacing in H&E stained breast WSIs, the closest level had 0.4861 μm pixel spacing; in H&E stained rectum WSIs, the closet level had 0.4549 μm; in PAS stained kidney WSIs, it had 0.5034 μm pixel spacing.

The average Dice scores of the thresholding method were between 0.8616 and 0.8778 and the sensitivities in the range from 0.5617 to 0.6745. It produced the highest false positive region count across all the tested levels, ranging from 20.83 to 38.33 average false positive regions per image.

The average Dice score of Otsu's method ranged from 0.5647 to 0.7865. It worked best on level 5. The average sensitivity was in the range of 0.2810 to 0.4450. The algorithm produced lower false positive counts than the thresholding method at the expense of significantly lower average Dice scores and Sensitivities.

The FESI algorithm achieved Dice scores between 0.8284 and 0.8419 and sensitivities between 0.5758 and 0.6495, outperforming Otsu's method in both metrics. The region selection method of the algorithm helped it achieve the lowest false positive region count across all tested levels: 12 and 9 false positives per image on levels 3 and 5, respectively.

The FCNN networks had the best average Dice score on all levels, ranging from 0.9775 to 0.9891, with the highest minimum and maximum and the lowest standard deviation across all levels compared to the baseline algorithms. For the complete list of metric values, we refer to Table 6.

In terms of slide area covered by false positive detections, false positives of FCNN A, B, and C accounted for 0.0207%, 0.0193%, and 0.0344% of the average slide area across levels 1, 3 and 5, respectively. The false positives of the thresholding method covered, 0.8594%, 0.8309%, and, 0.8057% on levels 1, 3 and 5, respectively. Otsu's method had 0.7268%, 0.5213%, and 0.5588%, while the FESI produced 0.0681%, and 0.0871% on the last two levels, respectively. Examples of segmentations are shown on Fig. 7.

**Table 6 Comparison of traditional methods and FCNNs.** The methods with the highest Dice score at each level are shown in bold.

| Method | Level | Pixel spacing (μm) | Dice score | | | | Sensitivity | | | | False positive count | | | |
|---|---|---|---|---|---|---|---|---|---|---|---|---|---|---|
| | | | Mean | Stdev | Min | Max | Mean | Stdev | Min | Max | Mean | Stdev | Min | Max |
| Thresholding | 1 | 0.5 | 0.8616 | 0.1346 | 0.4577 | 0.9898 | 0.5617 | 0.3586 | 0.0000 | 1.0000 | 38.33 | 73.37 | 0 | 387 |
| Otsu's | 1 | 0.5 | 0.5647 | 0.3680 | 0.0000 | 0.9769 | 0.2810 | 0.3783 | 0.0000 | 1.0000 | 28.57 | 37.30 | 0 | 176 |
| **FCNN A** | **1** | **0.5** | **0.9880** | **0.0089** | **0.9680** | **0.9970** | **0.9747** | **0.0637** | **0.7143** | **1.0000** | **6.93** | **9.46** | **0** | **34** |
| FCNN B | 1 | 0.5 | 0.9305 | 0.0740 | 0.7043 | 0.9942 | 0.6587 | 0.3396 | 0.0000 | 1.0000 | 4.03 | 4.69 | 0 | 18 |
| FCNN D | 1 | 0.5 | 0.9841 | 0.0132 | 0.9496 | 0.9958 | 0.9554 | 0.0820 | 0.7143 | 1.0000 | 6.77 | 10.06 | 0 | 44 |
| Thresholding | 3 | 2.0 | 0.8627 | 0.1361 | 0.4131 | 0.9905 | 0.5763 | 0.3631 | 0.0000 | 1.0000 | 34.50 | 76.49 | 0 | 412 |
| Otsu's | 3 | 2.0 | 0.7373 | 0.1596 | 0.2689 | 0.9523 | 0.2890 | 0.3665 | 0.0000 | 1.0000 | 12.63 | 12.09 | 0 | 54 |
| FESI | 3 | 2.0 | 0.8284 | 0.3288 | 0.0000 | 0.9951 | 0.6495 | 0.3690 | 0.0000 | 1.0000 | 1.37 | 2.63 | 0 | 12 |
| **FCNN B** | **3** | **2.0** | **0.9891** | **0.0085** | **0.9706** | **0.9968** | **0.9387** | **0.1111** | **0.6667** | **1.0000** | **4.00** | **5.93** | **0** | **25** |
| FCNN D | 3 | 2.0 | 0.9822 | 0.0195 | 0.9185 | 0.9961 | 0.8953 | 0.1302 | 0.5833 | 1.0000 | 5.37 | 8.25 | 0 | 31 |
| Thresholding | 5 | 8.0 | 0.8778 | 0.1293 | 0.4915 | 0.9924 | 0.6745 | 0.3631 | 0.0000 | 1.0000 | 20.83 | 29.65 | 1 | 167 |
| Otsu's | 5 | 8.0 | 0.7865 | 0.1623 | 0.3835 | 0.9690 | 0.4450 | 0.3903 | 0.0000 | 1.0000 | 12.17 | 10.90 | 0 | 48 |
| FESI | 5 | 8.0 | 0.8419 | 0.3105 | 0.0000 | 0.9912 | 0.5758 | 0.3619 | 0.0000 | 1.0000 | 0.57 | 1.69 | 0 | 9 |
| **FCNN C** | **5** | **8.0** | **0.9775** | **0.0186** | **0.9328** | **0.9945** | **0.6735** | **0.2762** | **0.2000** | **1.0000** | **1.77** | **2.75** | **0** | **11** |
| FCNN B | 5 | 8.0 | 0.9657 | 0.0284 | 0.8963 | 0.9946 | 0.7389 | 0.2248 | 0.3333 | 1.0000 | 1.10 | 2.18 | 0 | 11 |
| FCNN D | 5 | 8.0 | 0.9726 | 0.0253 | 0.9036 | 0.9951 | 0.7157 | 0.2383 | 0.2857 | 1.0000 | 1.70 | 2.04 | 0 | 7 |

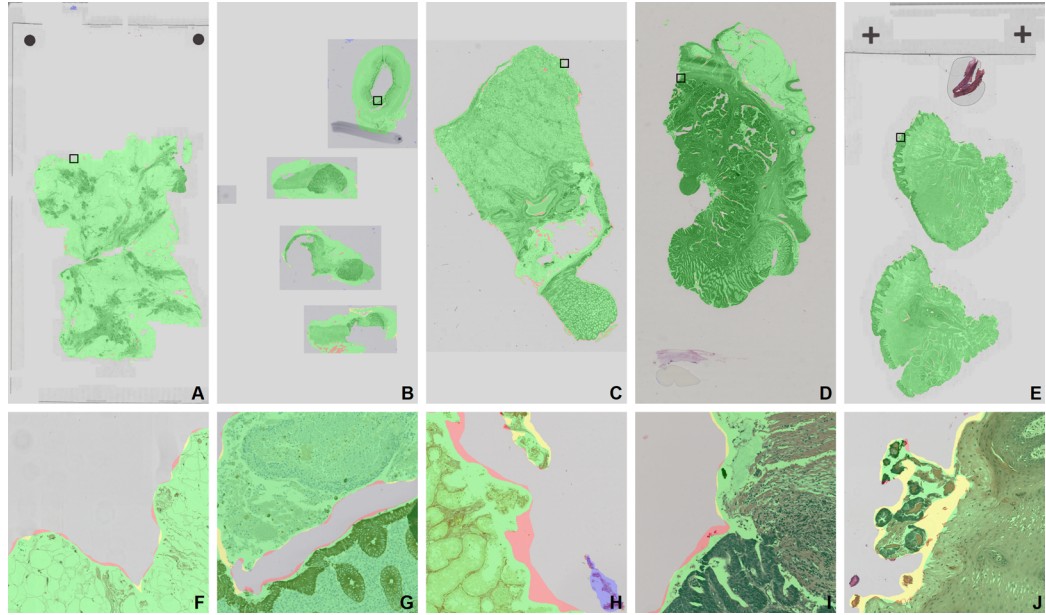

**Figure 7 Examples of segmentations by FCNN B from the development test set.** (A) and (F) Breast—H&E; (B) and (G) lymph node—CK8-18; (C) and (H) kidney—Sirius Red; (D) and (I) rectum—H&E; (E) and (J) tongue—H&E. Color overlays: correctly identified tissue is green, correctly identified background is gray, missed tissue is red, oversegmentation of tissue is yellow, false positive detections are blue. The zoomed areas of (F–J) are marked by black rectangles on the WSIs in (A)–(E). The field of view of the zoomed areas is 750 × 750 μm.

## Resolution-agnostic network

Next, we examined how the performance of an FCNN changes if we apply it on different resolution levels. We trained FCNN D with image patches extracted randomly from all the levels between 1 and 5. We compared it against the networks that were trained with a single pixel spacing on their native levels, and tested how a network trained on a single level performs at other levels.

The Dice score and sensitivity of the FCNN B trained on level 3 remained relatively high when tested on the two other levels: The 0.9891 average Dice score on its native pixel spacing only dropped to 0.9305 and 0.9657 on levels 1 and 5, respectively, while the average sensitivity dropped from 0.9968 to 0.9942 and 0.9946, respectively.

The FCNN D that was trained with image patches from levels 1, 3, and 5 achieved comparable average Dice scores and sensitivities to the FCNNs A, B, and C on all the tested levels; in average Dice score metrics, it outperformed FCNN B on its non-native levels: 1 and 5. FCNN D had a significantly higher average sensitivity on level 1 than on levels 3 and 5, and both FCNN B and FCNN D achieved better average sensitivities than FCNN C on level 5 (which FCNN C was exclusively trained on).

The resolution-agnostic FCNN D achieved similar false positive region counts to networks that were trained with a single pixel spacing across all the tested levels. The area covered by the false positive detections of FCNN D was 0.0299%, 0.0374%, and 0.0500% on levels 1, 3, and 5, respectively. Surprisingly, the FCNN B detects less false positive regions on levels 1 and 5 than the networks that were trained on that given level. For the complete list of achieved metrics, we refer to Table 6.

Furthermore, the paired $t$-tests calculated at level 5 revealed that the Dice score differences between FCNN D and the thresholding method, between FCNN D and Otsu's method, and between FCNN D and the FESI method were statistically significant ($p = 0.0007$, $p < 0.0001$, and $p = 0.0481$ respectively).

## Results on the dissimilar dataset

In order to test how well FCNN D performs on unfamiliar data, we tested it on the *dissimilar dataset* on level 3. Neither the eight images nor their tissue-staining categories were used for training or validation of the CNNs. While the *development dataset* did contain kidney and lung tissue WSIs, those were stained with PAS and Sirius Red and H&E, respectively.

FCNN D achieved an excellent average Dice score of 0.9828 and 0.9737 average sensitivity; only eight false positive regions were identified across all slides. Examples of segmentations are shown on Fig. 8. Full results are presented in Table 7.

## DISCUSSION

In this paper, we set out to build an algorithm to segment diagnostically-relevant tissue areas from WSIs across differing conditions, such as stains and scanners. The proposed CNNs outperformed the existing traditional methods across different resolution levels. FCNNs A, B, and C achieved significantly higher Dice scores and sensitivities than the simple thresholding method, Otsu's adaptive thresholding, and the FESI methods, with

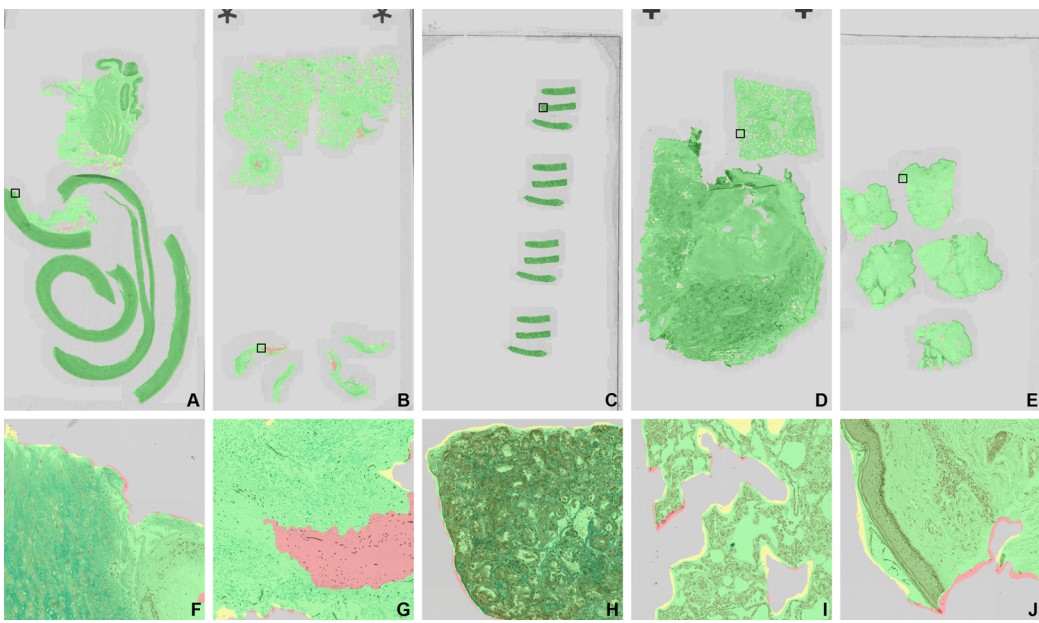

**Figure 8 Examples of segmentation by FCNN D from the dissimilar set.** (A) and (F) Aorta—Alcian Blue; (B) and (G) cornea—Grocott; (C and H) kidney—CAB; (D) and (I) lung—Perls; (E) and (J) skin—Von Kossa. Color overlays: correctly identified tissue is green, correctly identified background is gray, missed tissue is red, oversegmentation of tissue is yellow, false positive detections are blue. The zoomed areas of (F–J) are marked by black rectangles on the WSIs in (A)–(E). The field of view of the zoomed areas is 750 × 750 μm.

**Table 7 Results of FCNN D on the dissimilar dataset.** Since the dissimilar dataset had no tissue-staining category overlap with the development dataset, the results are good indicators of the generalization potential of the FCNNs.

| Tissue | Staining | Dice score | Sensitivity | False positive count |
|--------|----------|-----------|-------------|---------------------|
| Aorta | Alcian Blue | 0.9817 | 0.8333 | 1 |
| Brain | Alcian Blue | 0.9826 | 1 | 3 |
| Cornea | Grocott | 0.9449 | 0.9565 | 2 |
| Kidney | CAB | 0.9897 | 1 | 0 |
| Lung | Perls | 0.9920 | 1 | 0 |
| Skin | Perls | 0.9836 | 1 | 2 |
| Skin | Von Kossa | 0.9911 | 1 | 0 |
| Uterus | Von Kossa | 0.9980 | 1 | 0 |

only one exception on level 5. On level 5 the simple thresholding method achieved slightly higher sensitivity than FCNN C at the expense of lower Dice score and substantially higher false positive count. In addition, it was possible to train a resolution-agnostic network that performs equivalently to the networks that were trained on a single resolution level. FCNN D achieved Dice scores within 0.0069 of FCNs A, B, and C on their corresponding level. Finally, FCNN D showed excellent generalization potential, achieving 0.9828 average Dice score and 0.9737 average sensitivity at an average of one false positive per

image on the *dissimilar dataset*, demonstrating that it can perform well on unfamiliar tissue and staining types.

We did not include the published tissue segmentation method by *Hiary, Alomari & Chaudhary (2013)* in this study: They reported their results as a localization error in which a pathologist partly determined which errors were relevant (and thus counted), making the approach irreproducible for us. Re-implementing this method was not feasible due to missing algorithmic details.

FCNN B also worked well on its non-native levels, but the network did make several crucial errors. Most notably, it had problems recognizing fatty tissue on levels 1 and, to some extent, 5; these errors did not reduce the average Dice score significantly, but resulted in a substantially lower sensitivity. FCNN B achieved zero sensitivity on three of the test images and less than 0.5 on eight of the 30 test images on level 1. It also had difficulties with slightly out of focus-regions on both non-native levels. These problems were largely alleviated by the multi-level training of FCNN D. The mistakes of the two FCNNs were similar on levels 1 and 5, but the classification errors made by FCNN D were much smaller. Notably, FCNN D achieved better performances on finely-structured regions (e.g., lung tissue) on level 5 than FCNN C.

A drop in sensitivity from highest to lowest resolution levels was present across all methods, with an especially sharp drop from resolution level 3 to level 5 (see Table 6). This was mainly due to small annotated regions being missed. All networks had a 128 × 128 pixels training patch size, which means an increasing physical field of view from 64 and 256 μm to 1,024 μm from levels 1 to 5, respectively. The small tissue region with diameters slightly greater than 250 μm filled only a fraction of the training patch of the networks trained on level 5, while only 25% fit in the training patch of the networks trained on level 1. This made it easier for the network to neglect such regions on higher levels.

On the dissimilar dataset, FCNN D only missed two reference regions by falling below the 80% detection ratio minimum. It performed worse on the cornea tissue, with a Dice coefficient of 0.94. We hypothesize that this was caused by the texture of the lightly colored cornea tissue with the darkly stained fungi deposits; the appearance was very different from other tissues in the development dataset.

The proposed method can be a useful pre-processing step of any WSI processing algorithm. By accurately identifying the tissue areas it can effectively reduce the computational cost of subsequent processing steps. Integrating the proposed method into the software of whole-slide scanners is beyond the scope of this paper, but given the resolution-agnostic nature of FCNN D, one possible way would be to use the macro-scan of the complete glass slides (low resolution, results roughly similar to level 5 in Table 6), and let the algorithm find the relevant tissue regions with high sensitivity. Then the scanner could utilize the resulting tissue map, and scan the glass slide at the target resolution only where tissue was found. The algorithm can then during scanning be used at high-resolution to exclude artifacts and other non-tissue areas at high specificity, level 1 in Table 6.

This study has some limitations. While we collected images from a wide range of sources, our dataset is much too small to account for all variation encountered in the real world. Therefore, there are likely other tissue and staining types which the networks would

fail to identify. However, the results on the dissimilar dataset do indicate that a total segmentation failure is unlikely.

Furthermore, all of the slides in this study except the kidney CAB from the dissimilar dataset had four μm tissue thickness. We think that the wide variety of data in our study in terms of differing stains and tissue types already covers the variation one would encounter through different tissue thicknesses. Thinner tissue cuts produce lighter colored, less saturated images, while thicker cuts yields darker, more saturated images. These types of differences are also present in data from different centers with diverse scanners, or different stains and tissue types. Our augmentation pipeline included saturation and brightness adjustment steps with widened parameter range to accommodate this kind of variability. We hypothesize that the tissue thickness from the range of three to five μm should have no effect on the performance of the proposed method. This is further evidenced by the three μm thick tissue which shows similar results to the rest of the dataset.

## CONCLUSIONS

We would like to extend the study in the future by running it in a prospective setting at several centers. This would automatically expose it to more tissue types, stainings, tissue thicknesses, and scanners. By monitoring segmentation performance, we can identify potential failure modes and address them in a subsequent study. In this study, we did not compare our algorithms to the scanners' tissue-detection algorithms; this is another avenue for future research.

In conclusion, the FCNN solutions were proven to be superior compared to traditional methods. The resolution-agnostic FCNN variant was able produce excellent tissue-background segmentation across resolution levels, which allowed us to use a single algorithm for both coarse and fine segmentations. Lastly, we showed that this algorithm performed well on completely unfamiliar data.

## ACKNOWLEDGEMENTS

We thank for Dr. Filippo Fraggetta, Head of Pathology Department, Cannizzaro Hospital (Catania, Italy) for providing the example images of missed tissue regions shown in Fig. 1.

### Funding

This work was funded by the Automation in Medical Imaging project. Funding sources were Radboud University Medical Center and Fraunhofer-Gesellschaft. The funders had no role in study design, data collection and analysis, decision to publish, or preparation of the manuscript.

### Grant Disclosures

The following grant information was disclosed by the authors:
Automation in Medical Imaging project.
Radboud University Medical Center and Fraunhofer-Gesellschaft.

## Competing Interests

Jeroen van der Laak is a member of the scientific advisory boards of Philips, the Netherlands and ContextVision, Sweden and receives research funding from Philips, the Netherlands and Sectra, Sweden. Geert Litjens received research funding from Philips Digital Pathology Solutions (Best, the Netherlands) and has a consultancy role for Novartis (Basel, Switzerland).

## Author Contributions

- Péter Bándi conceived and designed the experiments, performed the experiments, analyzed the data, contributed reagents/materials/analysis tools, prepared figures and/or tables, authored or reviewed drafts of the paper, approved the final draft.
- Maschenka Balkenhol analyzed the data, authored or reviewed drafts of the paper, approved the final draft.
- Bram van Ginneken authored or reviewed drafts of the paper, approved the final draft.
- Jeroen van der Laak conceived and designed the experiments, authored or reviewed drafts of the paper, approved the final draft.
- Geert Litjens conceived and designed the experiments, analyzed the data, contributed reagents/materials/analysis tools, authored or reviewed drafts of the paper, approved the final draft.

## Human Ethics

The following information was supplied relating to ethical approvals (i.e., approving body and any reference numbers):

Institutional Review Board of the Radboud University Medical Center approved the study (2019-5161).

## Data Availability

The data is available at Zenodo: Bándi, Péter. (2019). Representative Sample Dataset for Resolution-Agnostic Tissue Segmentation in Whole-Slide Histopathology Images (Version 1.0.0) (Data set). Zenodo. DOI 10.5281/zenodo.3375528.

## Supplemental Information

Supplemental information for this article can be found online at http://dx.doi.org/10.7717/peerj.8242#supplemental-information.

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
