# Peer review of "Resolution-agnostic tissue segmentation in whole-slide histopathology images with convolutional neural networks"

_PeerJ, doi:10.7717/peerj.8242_

## Round 0.1 · original submission · Major Revisions

Both reviewers found your manuscript to be well-written. However, it is lacking in the area of algorithm description. Some statements are made, but not substantiated, for example, you need to detail how it deals with the missing-detection issue and dependence between the tissue thickness and performance. Comparison with the state-of-the-art approaches is also missing. I encourage you to modify your paper based on the reviews, provide point-by-point response to every comment and resubmit.

Reviewer 1 ·

Basic reporting

The reviewer agrees that the automated tissue detection in the scanning software has drawbacks, but the authors need to articulate them properly. Please provide specific examples showing circumstances that make scanners fail to detect tissues on slides. For instance, you can illustrate how the automated tissue detection works in Aperio and Pannoramic scanners. The statement made: “Unfortunately, there is no way to recover from errors made in tissue detection by slide scanners”, seems to be groundless. For instance, in user interface of Aperio scanning software, one can inspect and, if necessary, correct the bounding box drawn over the detected tissue. Once the tissue is inscribed in the bounding box, the scanner proceeds with scanning of area inside the bounding box. Software from other scanner manufacturers may have built in similar feature.

The authors say that “The typical file size of a WSI was around 4 GB, but it varied greatly depending on the scanner and tissue content of the image.” True, but this needs to be amended that the file size also depends on the scanning magnification (20x or 40x set manually in Aperio).

Example ground truth delineations need to be shown in a separate figure to give readers an idea on how they were made. Illustrate how experts annotated areas that have holes, normal lung parenchyma, or areas with adipocytes. Amend your manuscript with criteria your experts used while circling the tissue and separating it from background. These criteria are important to understand errors made by CNNs.

Image augmentation technique described in the manuscript seems to be overlapping in some parts with that described by Tellez at al (https://arxiv.org/pdf/1902.06543.pdf). Please list shared procedures, if there are any, and reduce the section volume if possible. Also, state what's new in your augmentation approach as compared to that by Tellez et al.

Inference section: explain the notion of the confidence value t. Is it the same as probability score outputted by CNN?

Ln 248. Explain what you meant by post-processing of measurements. Did you mean a post-processing of the resulting binary tissue mask?

Experimental design

Explain if and how a result obtained by CNN is affected by tissue thickness. A typical thickness of tissue re-cut for a histopathological evaluation is 4um. However, 3um or 5um sectioning is not uncommon. Include information about tissue sections included in the study. In If the slides (Table 2) are all 4um, the authors should collect slides that are 3um thick and analyze them with their CNNs and report performance. The performance should be evaluated statistically to identify potential differences arising from thicker vs. thinner tissue sections.

Validity of the findings

In introduction, the authors listed only most commonly used techniques for tissue detection. However, as shown for instance in https://www.ncbi.nlm.nih.gov/pubmed/25372389 or https://www.ncbi.nlm.nih.gov/pubmed/30728398 , a miniature RGB image of the whole WSI (say at the level equivalent to 5x) converted to a monochromatic image and then thresholded using a fixed intensity value seems to work sufficiently well to separate tissue from background. Please implement this simple technique, include its performance in Table 6, and discuss.

Methods section: clarify how exactly the Otsu reference method was applied to delineate tissue. WSI images are RGB matrices and the Otsu method works in principle on monochromatic images only.

Reviewer 2 ·

Basic reporting

This paper applied a fully convolutional neural network-based approach to segment/distinguish tissue versus non-tissue on tissue glass slides. It's overall well-structured and written. However, it lacks background introduction with enough reference and discussion of the state-of-the-art methods tackling the same question (the author mentioned two general segmentation methods, but how do people distinguish tissue versus background currently, how significant is this problem? Whether the reasonable missing of the tissue is acceptable in the clinic? The author should either provide literature evidence or self-generated data to support the current scanner has a high rate of failing to detect tissue and this really affect diagnostic accuracy).

Experimental design

The author applied a fully convolutional neural network to segment the tissue from the background on a pathology slide. The data and model were described with details.

However, the author mentioned in the introduction at line 48-50 the scanner could miss the tissue during the scanning. It looks the author aims to solve the missing-detection of the scanner. But I couldn't see how the proposed approach could solve this issue. Your approach and application are built upon after the image acquisition. Based on my understanding, the proposed resolution-agnostic network could scanner the at low-resolution to guide the scanner. But the author should explicitly describe how you would image happen and embed within the scanner itself. The fundamental transition and assumption should be provided.

Validity of the findings

The author provided a fully convolutional neural network-based approach to address the tissue segmentation problem on digitized tissue slides. The author compared with thresholding and structure detection approaches and stated the single FCN is multi-resolution enabled as well. Considering the previous evidence of deep learning-based approaches in image segmentation and classification, the result is not surprising. However, the author should state or compare why to choose the FCN rather U-Net or GAN based approach as they recently surpassed the FCN.

Additional comments

The overall paper is well-written with supportive table and data. However, the author should provide more background with reference/own data regarding why the problem addressing is significant. Also, the author should compare with more state-of-the-art approaches.

---

## Round 0.2 · accepted · Accept

Thank you for addressing the concerns raised by the reviewers. The manuscript is now technically sound. It may be a good idea to upload your code on Github.

Reviewer 1 ·

Basic reporting

No comment

Experimental design

No comment

Validity of the findings

no comment

Additional comments

Thank you. The authors addressed all my comments properly.

Reviewer 2 ·

Basic reporting

The author has sufficiently addressed previous issues.

Experimental design

The author has sufficiently addressed previous issues.

Validity of the findings

The author has sufficiently addressed previous issues.

Additional comments

The author has sufficiently addressed previous issues.